# Disentangling a Thorny Issue: Myocarditis and Pericarditis Post COVID-19 and Following mRNA COVID-19 Vaccines

**DOI:** 10.3390/ph15050525

**Published:** 2022-04-25

**Authors:** Concetta Rafaniello, Mario Gaio, Alessia Zinzi, Maria Giuseppa Sullo, Valerio Liguori, Marialuisa Ferraro, Fiorella Petronzelli, Patrizia Felicetti, Pasquale Marchione, Anna Rosa Marra, Francesco Rossi, Antonella De Angelis, Annalisa Capuano

**Affiliations:** 1Campania Regional Centre for Pharmacovigilance and Pharmacoepidemiology, 80138 Naples, Italy; concetta.rafaniello@unicampania.it (C.R.); mario.gaio@unicampania.it (M.G.); pina.sullo@libero.it (M.G.S.); valerio93liguori@gmail.com (V.L.); marialuisaferraroo@gmail.com (M.F.); francesco.rossi@unicampania.it (F.R.); annalisa.capuano@unicampania.it (A.C.); 2Section of Pharmacology “L. Donatelli”, Department of Experimental Medicine, University of Campania “Luigi Vanvitelli”, 80138 Naples, Italy; antonella.deangelis@unicampania.it; 3Pharmacovigilance Unit, Post-Marketing Surveillance Area, Italian Medicine Agency, 00187 Rome, Italy; f.petronzelli@aifa.gov.it; 4Post-Marketing Surveillance Area, Italian Medicine Agency, 00187 Rome, Italy; p.felicetti@aifa.gov.it (P.F.); ar.marra@aifa.gov.it (A.R.M.); 5Signal Management Unit, Post-Marketing Surveillance Area, Italian Medicine Agency, 00187 Rome, Italy; p.marchione@aifa.gov.it

**Keywords:** VAERS, myocarditis, pericarditis, myopericarditis, COVID-19 vaccines, mRNA vaccines, safety monitoring

## Abstract

Considering the clinical significance for myocarditis and pericarditis after immunization with mRNA COVID-19 vaccines, the present pharmacovigilance study aimed to describe these events reported with mRNA COVID-19 vaccines in the Vaccine Adverse Events Reporting System (VAERS). From 1990 to July 2021, the mRNA vaccines were the most common suspected vaccines related to suspected cases of myocarditis and/or pericarditis (myocarditis: N = 1,165; 64.0%; pericarditis: N = 743; 55.1%), followed by smallpox vaccines (myocarditis: N = 222; 12.2%; pericarditis: N = 200; 14.8%). We assessed all suspected cases through the case definition and classification of the Brighton Collaboration Group, and only definitive, probable, and possible cases were included in the analysis. Our findings suggested that myocarditis and pericarditis mostly involve young male, especially after the second dose with a brief time to onset. Nevertheless, this risk is lower (0.38/100,000 vaccinated people; 95% CI 0.36–0.40) than the risk of developing myocarditis after SARS-CoV-2 infection (1000–4000 per 100,000 people) and the risk of developing “common” viral myocarditis (1–10 per 100,000 people/year). Comparing with the smallpox vaccine, for which is already well known the association with myocarditis and pericarditis, our analysis showed a lower probability of reporting myocarditis (ROR 0.12, 95% CI 0.10–0.14) and pericarditis (ROR 0.06, 95% CI 0.05–0.08) following immunization with mRNA COVID-19 vaccines.

## 1. Introduction

During the nationwide vaccination campaign from December 2020 through May 2021, the Israeli Ministry of Health recorded 136 cases of definite or probable myocarditis that had occurred with a temporal relationship with the receipt of two doses of the BNT162b2 mRNA vaccine [1]. At the same time, in April 2021, the European Medicines Agency’s (EMA) Pharmacovigilance Risk Assessment Committee (PRAC) started the review of suspected myocarditis and pericarditis reported after immunization with the mRNA COVID-19 vaccines (Pfizer-BioNTech and Moderna) in both Israel and European Economic Area [2]. Myocarditis and pericarditis are inflammatory diseases involving the myocardium and the pericardium that can occur following systemic conditions, such as malignancy, inflammatory responses, or autoimmune disorders [3,4]. Viral infections are a known common cause of myocarditis and pericarditis, due to a combination of direct cellular injury and immune-mediated cytotoxicity. Adenoviruses, enteroviruses, parvovirus B19, and the Herpesviridae family as well as human immunodeficiency virus, hepatitis C virus, influenza A virus and influenza B virus, and, also, Coronavirus have been associated to inflammatory diseases of the heart, including Middle East respiratory syndrome coronavirus (MERS-CoV), severe acute respiratory syndrome coronavirus (SARS-CoV), and SARS-CoV-2, that share a common angiotensin-converting enzyme 2 (ACE2) tropism, potentially mediating both direct and indirect cardiac damage, such as cytokine-mediated cardiotoxicity or autoimmune response [5,6].

Although myocarditis and pericarditis may affect all age groups, ethnicities, and both genders, young and middle-aged male adults are more frequently affected, with the median age at diagnosis being 42 years old [7]. The clinical presentation of both diseases is overlapping and can include shortness of breath, chest pain, and abnormal heart rhythm as well as other signs and symptoms of a viral infection such as a headache, body aches, joint pain, fever, a sore throat, or diarrhea [8,9]. Currently, diagnosis of myocarditis is predominantly based on cardiac magnetic resonance imaging (cMRI) in patients with classical clinical features associated to an otherwise unexplained elevation in troponin, while pericarditis on both computed tomography (CT) and magnetic resonance (MR). Transthoracic echocardiography (TTE), electrocardiogram (EKG), and chest X-ray (CXRk) may also be helpful [8,9].

Overall, incidence and prevalence of myocarditis are considered to be significantly underestimated. In a recent study using International Classification of Diseases (9th revision) codes, the global prevalence of myocarditis has been estimated to be ≈22 of 100,000 patients per year [10]. Similarly, exact epidemiological features for acute pericarditis are still lacking. The incidence of acute pericarditis is approximately 27.7 per 100,000 individuals in the Western world, whereas few data are available from low-income countries, where tuberculous etiology is the leading cause of acute pericarditis [11]. 

Although myocarditis has been reported as a rare adverse event following other vaccines, smallpox vaccination is the only vaccine that has ever been conclusively linked to myocarditis based on a significantly higher relative risk [12,13]. Meanwhile, on the basis of data from the Vaccine Adverse Events Reporting System (VAERS), the Centers for Disease Control and Prevention (CDC) has estimated that the incidence of myocarditis after any COVID-19 vaccination is 0.48 cases per 100,000 overall and 1.2 cases per 100,000 among vaccine recipients between the ages of 18 and 29 years [14].

In this regard, considering the clinical significance for myocarditis and pericarditis after the administration of mRNA COVID-19 vaccines, the present pharmacovigilance study aimed to describe these events reported for mRNA COVID-19 vaccines in the VAERS database. We also aimed to assess the reporting odds ratio (ROR) and 95% confidence interval (CI) to evaluate if mRNA COVID-19 vaccines have a lower/higher probability of reporting Individual Case Safety Reports (ICSRs) with myocarditis or pericarditis in a direct comparison with smallpox vaccine and compared with all other vaccines.

## 2. Results

From the 1 January 1990 to the 20 July 2021, 1,048,575 AEFI-reporting ICSRs were retrieved from VAERS, of which 2772 (0.26%) included myocarditis and/or pericarditis as an adverse event. Of these, 1552 ICSRs were suspected cases of myocarditis, 1164 ICSRs reported suspected cases of pericarditis, and 56 ICSRs included both myocarditis and pericarditis (defined as myopericarditis in this study). Given that an ICSR could include more than one suspected vaccine, 1821, 1348, and 100 vaccines were reported as suspected in ICSRs including suspected cases of myocarditis, pericarditis, and myopericarditis, respectively. Regarding ICSRs which described myocarditis or pericarditis as AEFI, mRNA vaccines were the most common suspected vaccine (myocarditis: N = 1165; 64.0%; pericarditis: N = 743; 55.1%), followed by smallpox vaccines (myocarditis: N = 222; 12.2%; pericarditis: N = 200; 14.8%), and anthrax vaccines (myocarditis: N = 63; 3.5%; pericarditis: N = 63; 4.7%). In the examined period, no mRNA vaccines were reported as suspected in those ICSRs related to suspected cases of myopericarditis, while smallpox vaccines were the most common in myopericarditis ICSRs (N = 51; 51.0%), followed by anthrax vaccines (N = 23; 23.0%).

Through the application of the algorithm drawn up by the Brighton Collaboration, 1063 cases of myocarditis, 741 cases of pericarditis, and 37 cases of myopericarditis were identified and included in the study. Relatively to the suspected cases of myocarditis, we classified 201 cases as “Definitive” (Level 1), 854 cases as “Probable” (Level 2), and 8 cases as “Possible” (Level 3); the remaining ICSRs (N = 489) were excluded from the study because assessed as Level 4 (N = 462) and Level 5 (N = 27). Similarly, the cases of pericarditis evaluated as “Definitive” (N = 449), “Probable” (N = 239), and “possible” (N = 53) were included in the study, while the remaining 423 ICSRs were excluded because assessed as Level 4 (N = 260) and Level 5 (N = 163). Finally, 30 cases of myopericarditis have been evaluated as “Definitive”, 6 cases as “Probable”, and 1 case as “Possible”. The remaining 19 cases were assessed as Level 4 (N = 17) and Level 5 (N = 2) (Figure 1). The inclusion criteria are extensively described above.

As stated above, considering that one ICSR could include more than one suspected vaccine, among the 1063 included ICSRs of myocarditis, 1267 vaccines were reported as suspected; mRNA COVID-19 vaccines were the most reported (N = 778; 61.4%), followed by the smallpox vaccines (N = 281; 22.2%), anthrax vaccines (N = 57; 4.5%), and influenza vaccines (N = 39; 3.1%).

Regarding to the 741 ICSRs of pericarditis included in the study, 864 vaccines were reported as suspected, and the 54.6% (N = 472) of which was an mRNA COVID-19 vaccine, followed by the smallpox vaccines (N = 151; 17.5%), the influenza vaccines (N = 44; 5.1%) and the anthrax vaccines (N = 43; 5.0%). The smallpox vaccines were the most common ones (N = 33; 51.6%) reported as suspected among ICSRs describing cases of myopericarditis, followed by the anthrax vaccines (N = 13; 20.3%), and the influenza vaccines (N = 2; 3.1%), while no mRNA vaccines were reported as suspected (Figure 2).

Looking at the ICSRs related to mRNA COVID-19 vaccines with myocarditis as AEFI (N = 778), our analysis showed that this event was more commonly reported for male subjects (N = 626; 80.5%) aged between 12 and 40 years (N = 650; 83.5%). Analogously, cases of pericarditis following mRNA vaccines (N = 472) were mainly observed in male subjects (N = 336; 71.2%) aged between 12 and 40 years (N = 277; 58.7%). The hospitalization was the most frequent outcome both for myocarditis (N = 640; 81.1%) and pericarditis (N = 268; 76.1%); the average length of hospitalization was 3 days both in cases of myocarditis and pericarditis (Table 1).

More than half of the cases involved events after the second dose of vaccination (N = 751; 60.1%). Regarding to myocarditis, it mainly occurred between the first and the third day after the second dose of vaccination; regarding to pericarditis, it mainly occurred between the first and the 13th day after the second dose of vaccination with a median time to onset of 2 days from the second dose administration for myocarditis and 3 for pericarditis (Figure 3). Additionally noteworthy is the spreading of TTO in myocarditis and pericarditis following the first dose administration compared to the second one for which the cases are more clustered.

As shown in Figure 4, the frequency of cases reported decreases with increasing age: from 476 cases in the 12–20 years age group (38.1%) to ≤5% of cases in vaccinees above 61 years of age (age was not available for 36 ICSRs).

Most cases of myocarditis and pericarditis following COVID-19 immunization with mRNA vaccines were reported after the second dose (N = 751; 60.1%), and among these, 42.9% (N = 322) were related to subjects aged between 12 and 20 years, and the number of reports decreased with increasing age (Figure 5).

From disproportionality analysis, mRNA COVID-19 vaccines were associated to a higher risk of reporting myocarditis (ROR = 3.76, 95% CI 3.45–4.09, *p* << 0.001) and pericarditis (ROR = 2.27, 95% CI 2.05–2.52, *p* << 0.001) compared to other vaccines. These results are confirmed in all age group for cases of myocarditis (12–50 years age group: ROR = 6.49, CI 5.88–7.16, *p* << 0.001; over 50 years age group: ROR = 4.13, CI 2.95–5.72, *p* << 0.001), and for cases of pericarditis (12–50 years age group: ROR = 4.27, CI 3.72–4.89, *p* << 0.001; over 50 years age group: ROR = 5.88, CI 4.73–7.29, *p* << 0.001). 

On the contrary, in both the general population and for each age group, mRNA COVID-19 vaccines were associated to a lower risk of reporting cases of myocarditis and pericarditis compared to smallpox vaccine (myocarditis: ROR 0.12, 95% CI 0.10–0.14, *p* << 0.001, in the general population; ROR 0.11, 95% CI 0.09–0.12, *p* << 0.001, in the 12–50 years age group; ROR 0.06, 95% CI 0.02–0.52, *p* = 0.007, in the over 50 years age group; pericarditis: ROR 0.06, 95% CI 0.05–0.08, *p* << 0.001, in the general population; ROR 0.05, 95% CI 0.04–0.07, *p* << 0.001, in the 12–50 years age group; ROR 0.08, 95% CI 0.03–0.29, *p* = 0.003, in the over 50 years age group) (Figure 6).

## 3. Discussion

In this study, from 1 January 1990 to 20 July 2021, we analyzed all ICSRs with myocarditis, pericarditis, and myopericarditis reported after vaccines’ administration through the Vaccine Adverse Event Reporting System (VAERS), the US national vaccine safety passive monitoring system database. In order to provide a descriptive analysis of these specific cardiac events, we included all cases which met the case definition and classification of the Brighton Collaboration for myocarditis and pericarditis [15]. During the study period, 1,048,575 ICSRs were analyzed and 1841 met the case definition of the Brighton Collaboration for myocarditis and pericarditis. Specifically, N.1063 cases of myocarditis, N.741 cases of pericarditis and N.37 cases of myopericarditis have been identified and overall, most of these events, 67.9%, were associated to mRNA COVID-19 vaccines. Moreover, looking at the demographic characteristics, most of the identified cases were reported for male and young patients and after the second dose of vaccination. These results are consistent with the already available evidence about myocarditis and pericarditis in mRNA vaccine recipients issued by regulatory authorities [16,17], and well described in several observational studies. Indeed, in this regard, considering the most recent retrospective patient studies which similarly to our analysis have used the case definition and classification of the Brighton Collaboration for myocarditis and pericarditis, these cardiac adverse events have been observed mainly in male patients between the ages of 16 and 29 years; moreover, most of the events have occurred after the second dose of mRNA vaccines [1,18]. The present evaluation is in line with these conclusions; however, the risks might be underestimated due to the fact that the younger age groups (below 40 years old) fully vaccinated for COVID-19 were still under the 50% population coverage in the examined period. To date, the underline mechanism of myocarditis following mRNA immunization has not been fully defined; however, in general, it is well known that pathogens, viruses included, could lead to these inflammatory cardiac disorders as well as some drugs, vaccines, and systemic immunological diseases [5]. As suggested by Montgomery J et al., just considering the clinical course of the 23 patients with clinical evidence of myocarditis following mRNA COVID-19 vaccination, this event could be better defined as eosinophilic hypersensitivity myocarditis and to support this definition, the authors have considered that most of the analyzed cases, similarly to our findings, occurred after the second dose, thus suggesting a prior exposure, but also the brief time to onset [18]. Moreover, a recent study, through the application of a systems biology informatics approach to the VAERS data, has suggested that IFN-gamma signaling, an immune effector for several vaccines, and TH1 immune responses could play an important role on the postvaccine-myocarditis, bringing into play an autoimmune mechanism [19]. Additionally, this proposed mechanism could also explain the age distribution among myocarditis following mRNA COVID-19 vaccines, in fact, it is well recognized that the levels of proinflammatory cytokines increase at puberty [19]; in parallel, as it was suggested for the different gender distribution among patients with severe COVID-19 disease, also in this case, the hypothetic estrogen anti-inflammatory effect or its vascular and cardiac protective role could explain the prevalence of male gender for postvaccine-myocarditis [20]. Similarly, the autoimmune phenomenon has been proposed as the underlying mechanism of myocarditis and pericarditis following smallpox vaccination. More in detail, it has been suggested that the viral antigen could mimic the proteins of the myocardium, thus making the immune system unable to recognize the first one [21]. Indeed, smallpox vaccine is known to have the strongest association with these events based on a significantly higher relative risk compared to other vaccines [22,23]. Based on that, disproportionality analysis was performed in order to assess the probability of reporting ICSRs with myocarditis and pericarditis for mRNA COVID-19 vaccines compared to smallpox vaccine as well as all other vaccines. Our findings suggested a higher reporting risk associated to mRNA COVID-19 vaccines, both in the general population and in each considered age category. On the contrary, comparing mRNA COVID-19 vs. smallpox vaccines our analysis showed a ROR <1 (*p* < 0.01), suggesting that the former is associated to a lower risk of reporting myo/pericarditis compared to the latter. If, on one hand, these results could suggest that myo- and pericarditis are less observed following mRNA COVID-19 vaccines compared with the smallpox one, on the other, they could be, also, due to the fact that less severe cases of myo- and pericarditis could have not been reported or even not diagnosed. In this regard, looking at our results, most of the identified cases were associated to hospitalization, thus suggesting a high severity of the reported events. However, we cannot rule out the potential role of independent predictors factors such as autoimmune disorders as well as systemic inflammatory conditions or beta-blocker therapy in the occurrence of life-threatening and fatal myocarditis and/or pericarditis. Indeed, Engler RJ et al., in order to determine the prospective incidence of new onset of cardiac symptoms and subclinical myocarditis and pericarditis following immunization with either smallpox or trivalent influenza vaccine, have demonstrated that 60% of the observed cases would not have been diagnosed outside of the study cohort [23]. Similarly, our evaluation is based on spontaneous reporting that, as known, might be lacking valuable clinical information. Having used the case definition and classification of the Brighton Collaboration to select the study events, we cannot rule out the possibility that some potential use cases of myo- and pericarditis were not included in the present study. Indeed, the unprecedented public interest in the new COVID-19 vaccines has led to an unexpected increase in spontaneous reports of suspected adverse events often of poor quality which can seriously hamper with the detection of new signals.

## 4. Materials and Methods

### 4.1. Data Source

Data on Individual Case Safety Reports (ICSRs) were retrieved from the Vaccine Adverse Event Reporting System (VAERS). This database was created by the Food and Drug Administration (FDA) and Centers for Disease Control and Prevention (CDC) to receive reports about suspected Adverse Events Following Immunization (AEFI) [24]. We searched VAERS reports submitted from 1990 to 20 July 2021 following all vaccines irrespective of any variables such as race/ethnicity. From each ICSR the following informations were considered: (i) the patient (i.e., gender, age, co-medications, illness at time of vaccination, chronic or long-standing health conditions, and allergies to medications, food, or other products), (ii) the vaccine (i.e., vaccine type, vaccine manufacturer, manufacturer’s vaccine batch number, number of doses administrated, route administration, vaccination site, vaccine name, vaccination date, and prior vaccination event information), and (iii) the AEFIs (i.e., description, date of onset, outcome, and diagnostic laboratory data). With respect to the AEFIs, they are coded as Preferred Term (PT) according to the Medical Dictionary for Regulatory Activities (MedDRA) [25]. These data were originally included in three different datasets: (i) “VAERSDATA”, which contains data on patients and about the outcome of the AEFIs, with their related description (narrative field), and the laboratory/diagnostic data, (ii) “VAERSVAX”, which provides the vaccine information, and (iii) “VAERSSYMPTOMS”, which provides the AEFIs coded according to the MedDRA dictionary. The ICSRs’ identification codes have been used as a unique key between the three above mentioned datasets to build our own dataset. Moreover, through the analysis of the narrative fields, it was also possible to retrieve additional valuable information such as patient medical history, concomitant medication information, clinical course of the adverse events, seriousness and outcome of the events, laboratory findings, action taken, and dechallenge and rechallenge information (when available).

### 4.2. Case Assessment

VAERS data are from a passive surveillance system, and it is important to note that for any reported event, no cause-and-effect relationship has been established. So, after selecting all spontaneously reported cases of myocarditis and pericarditis following vaccination, we assessed them through the case definition and classification of the Brighton Collaboration [15] in order to minimize the uncertainty of the selected cases. This case definition was developed by a group of experts in the context of active development of vaccines for COVID-19 and other emerging pathogens. The definitions have been formulated with five levels of certainty: Level 1 (Definitive case), which is highly specific for the identification of a case of myocarditis and pericarditis, Level 2 (Probable case), and Level 3 (Possible case).

Regarding to the myocarditis case definition, the Level 1 classification can be obtained either by a myocardial inflammation showed by a histopathologic examination or by a combination of elevated myocardial biomarkers with an abnormal imaging study (either cardiac magnetic resonance (CMR) or echocardiography). The Level 1 classification does not require symptomatology because it was assumed that decisions to test for elevated myocardial biomarkers, CMR, or echocardiography would be driven by symptoms of myocarditis. The Level 2 can be reached by the presence of clinical symptoms and at least one abnormal CMR, electrocardiogram, echocardiogram, or elevated cardiac biomarker test result. The Level 3 requires the presence of clinical symptoms and abnormal inflammatory markers or an electrocardiogram without the characteristic findings of myocarditis. The symptoms include cardiac symptoms, such as acute chest pain, palpitations, dyspnea after exercise, diaphoresis, and sudden death, and non-specific symptoms, such as fatigue, abdominal pain, dizziness, edema, and cough. Finally, the Level 4 represents a reported event of myocarditis but with insufficient evidence to meet Level 1, 2, or 3 of the case definition, and Level 5 is a non-case.

Regarding to the pericarditis case definition, Level 1 classification can be reached either by observation pericardial inflammation by a pericardial biopsy or autopsy, or at least two abnormal results of the following: (i) evidence of abnormal fluid collection or pericardial inflammation determined by imaging, (ii) abnormalities showed by electrocardiogram that are new, or (iii) characteristic physical examination findings for pericarditis. The Level 2 requires clinical symptoms (at least 1 of the following: acute chest pain, palpitations, dyspnea after exercise, diaphoresis, and sudden death) and physical examination findings or imaging suggestive of an abnormal fluid collection or abnormal findings on electrocardiogram. The Level 3 includes cases with non-specific symptoms and at least one of the following results from instrumental exams: enlarged heart on chest X-ray, non-specific electrocardiogram abnormalities that are new and/or normalize on recovery. The Level 4 is a reported case of pericarditis that fails to meet Level 1, 2, or 3 of the case definition, and Level 5 is a non-case. 

Only the first three levels of certainty (“definitive”, “probable”, and “possible” cases) have been included in the present study.

### 4.3. Statistical Analysis

We performed a descriptive analysis using data of myocarditis and pericarditis following mRNA COVID-19 vaccines, obtaining the following information: the total number of cases, the number of cases split for gender, the number of cases for different age groups (12–20 years, 21–30 years, 31–40 years, 41–50 years, 51–60 years, 61–70 years, 71–80 years, and 80+ years), and the median age. Moreover, we reported the data on the dose number administered, and the outcome (classified as “Fatal”, “Life-threatening”, “Disability”, and “Hospitalization”). Regarding to the cases including the outcome “hospitalization”, it was also considered the hospital stay (number of days hospitalized), when available. Moreover, we assessed the median time to event onset from both the first and the second dose. Finally, we used the reporting odds ratio (ROR) with a 95% of confidence interval (95% CI) to investigate disproportional reporting of myocarditis and pericarditis between mRNA COVID-19 vaccines and other vaccines, and between mRNA COVID-19 vaccines and smallpox vaccine. All statistical analysis was performed using R Statistical Software (version 4.0.3; R Foundation for Statistical Computing, Wien, Austria).

## 5. Conclusions

Our findings, in line with other recent studies, highlighted that mRNA COVID-19 vaccines could be potentially associated to myocarditis and pericarditis as already have been pointed out by regulatory authorities; such events mostly involve young male especially after second dose with a brief time to onset. Nevertheless, according to the existing evidence, this risk is lower than the risk of developing myocarditis and cardiac injury after SARS-CoV-2 infection (1000–4000 per 100,000 people) and the risk of developing “common” viral myocarditis (1–10 per 100,000 people/year); indeed, our analysis has shown an incidence of myocarditis following COVID-19 mRNA vaccination equal to 0.38 per 100,000 vaccinated people (95% CI 0.36–0.40), in line with previous literature (0.3–0.5 per 100,000 vaccinated people) [26]. Regarding to the severity of myocarditis, the current literature suggests that hospitalized patients diagnosed with myocarditis following mRNA COVID-19 vaccines survived and recovered cardiac function within 1–5 weeks after an initial hospitalization of 3–5 days, comparing to the same event following SARS-CoV-2 infection (survival rate: >99% vs. 30–80%).

Comparing with the smallpox vaccine, for which is already well known the association with myocarditis and pericarditis, our analysis showed a lower probability of reporting such events following immunization with mRNA COVID-19 vaccines. 

In this regard, we want to point out that reporting odds ratio method for disproportionality analysis applied to pharmacovigilance databases could provide useful data, despite the already well-known weakness compared to renewed risk measures of etiological studies; indeed, our results not only suggest that comparing to smallpox vaccine, for which the risk of myocarditis is already known, mRNA COVID-19 vaccines are associated to a lower probability of reporting that event, but that the clinical characteristics of the selected cases are in line with those described today in the scientific literature. As suggested by several observational studies, myocarditis and pericarditis following immunization with mRNA COVID-19 vaccines are characterized by a signs, symptoms, and clinical course that resemble an immunoinflammatory response to vaccination. Although the underline mechanism has not been established yet, recently both myocarditis and pericarditis have been added in the Summary of the Product Characteristics of the available mRNA COVID-19 vaccines. However, further confirmatory studies are needed in order to verify a causal association between mRNA COVID-19 vaccines and both myocarditis and pericarditis. Finally, concerns about rare adverse events following immunization in general should not negatively affect overall trust in the value of this key component of primary health care.

## Figures and Tables

**Figure 1 pharmaceuticals-15-00525-f001:**
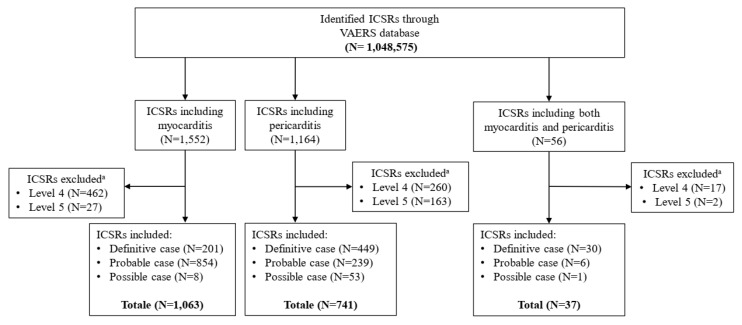
Flowchart for Individual Case Safety Reports inclusion and exclusion criteria. ^a^ ICSRs have been classified as Level 4 in case of reported events with insufficient evidence to meet Level 1 (Definitive case), Level 2 (Probable case), and Level 3 (Possible case); Level 5 is a non-case of myocarditis, pericarditis, and myopericarditis.

**Figure 2 pharmaceuticals-15-00525-f002:**
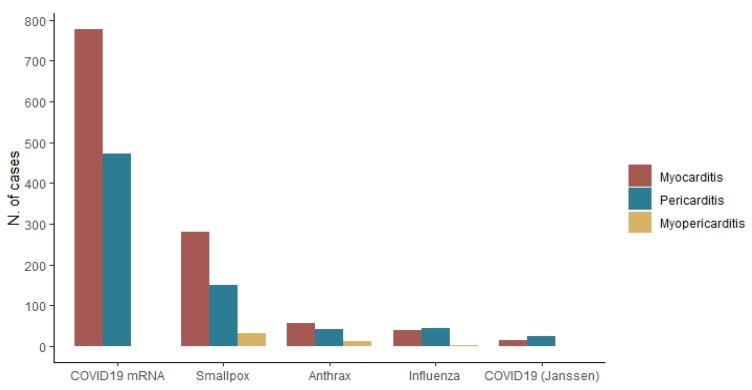
Distribution of myocarditis (n = 1063), pericarditis (n = 741), and myopericarditis (n = 37) by each vaccine for which study events have been reported.

**Figure 3 pharmaceuticals-15-00525-f003:**
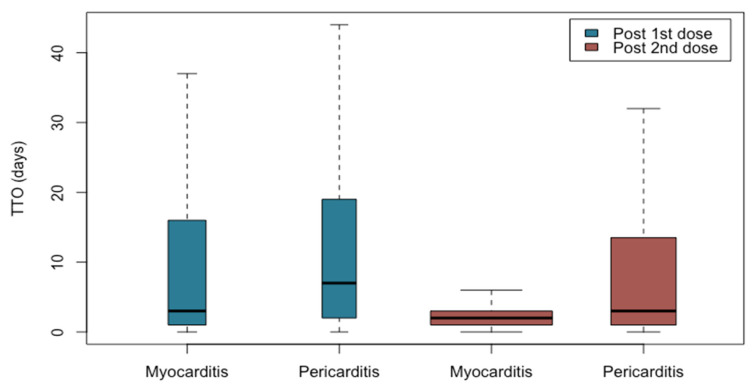
Boxplot—time to onset of myocarditis and pericarditis followed by the first and second dose of mRNA COVID-19 vaccines.

**Figure 4 pharmaceuticals-15-00525-f004:**
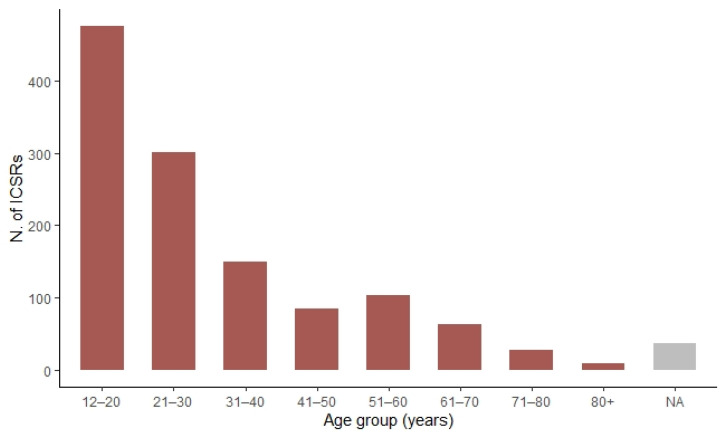
Distribution of cases of myocarditis and pericarditis following mRNA COVID-19 vaccines with stratification by age group.

**Figure 5 pharmaceuticals-15-00525-f005:**
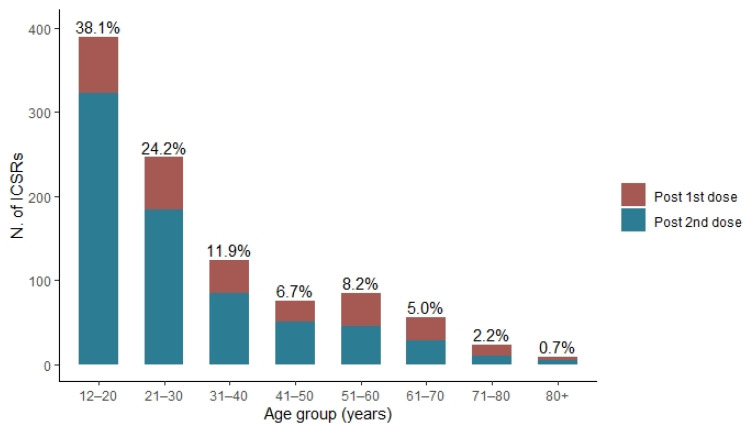
Distribution of cases of myocarditis and pericarditis following the first and second dose of mRNA COVID-19 vaccines with stratification by age group.

**Figure 6 pharmaceuticals-15-00525-f006:**
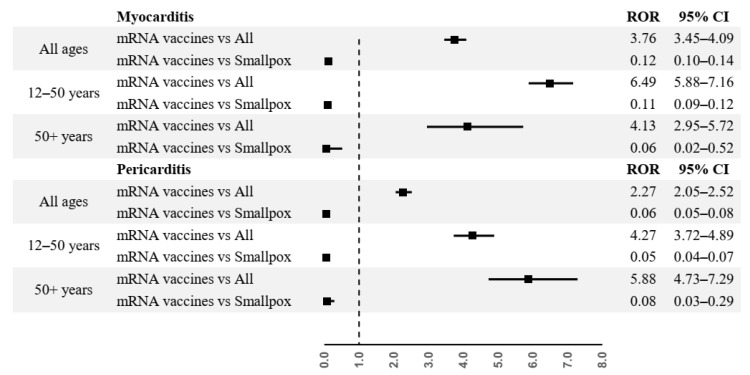
Disproportionality analysis.

**Table 1 pharmaceuticals-15-00525-t001:** Demographic characteristics of ICSRs reporting cases of myocarditis and pericarditis followed by COVID-19 immunization with mRNA vaccines. The outcomes are not mutually exclusive.

	Myocarditis N = 778 (%)	Pericarditis N = 472 (%)
**Gender**		
Male	626 (80.5)	336 (71.2)
Female	145 (18.6)	135 (28.6)
NA	7 (0.9)	1 (0.2)
**Age group**		
12–20 years	360 (46.3)	116 (24.6)
21–30 years	208 (26.7)	94 (19.9)
31–40 years	82 (10.5)	67 (14.2)
41–50 years	44 (5.7)	40 (8.5)
51–60 years	34 (4.4)	69 (14.6)
61–70 years	17 (2.2)	46 (9.7)
71–80 years	5 (0.6)	23 (4.9)
80+ years	1 (0.1)	8 (1.7)
NA	27 (3.5)	9 (1.9)
**Median age (IQR ^a^)**	21 (16–30)	34 (20–55)
**Outcome**		
Fatal	6 (0.8)	1 (0.3)
Life threatening	126 (16.0)	67 (19.1)
Disability	17 (2.1)	16 (4.5)
Hospitalization	640 (81.1)	268 (76.1)
Mean hospitalization time in days (±SD ^b^)	3 (±1.7)	3 (±2.1)
**Dose**		
Post 1st dose	152 (19.5)	137 (29.0)
Post 2nd dose	489 (62.9)	262 (55.5)
NA	137 (17.6)	73 (15.5)
**Median TTO ^c^** **post 1st dose (IQR)**	3 (1–16)	7 (2–19)
**Median TTO ^c^** **post 2nd dose (IQR)**	2 (1–3)	3 (1–13)

^a^ IQR = interquartile range; ^b^ SD = standard deviation; ^c^ TTO = time to onset.

## Data Availability

Publicly available datasets were analyzed in this study. These data can be found here: https://vaers.hhs.gov/ (accessed on 20 July 2021).

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
