# Peer review of "Disentangling a Thorny Issue: Myocarditis and Pericarditis Post COVID-19 and Following mRNA COVID-19 Vaccines"

_pharmaceuticals, 2022, doi:10.3390/ph15050525_

Round 1
Reviewer 1 Report
Rafaniello et al sought to investigate the myocarditis and pericarditis events reported with mRNA COVID-19 vaccines in the Vaccine Adverse Events Reporting System. The authors found that myocarditis and pericarditis mostly involve young male, especially after the second dose with a brief time to onset but its risk is lower than the risk of developing myocarditis after SARS-CoV-2 infection and the risk of developing “common” viral myocarditis. This analysis is very important in terms of public health. Some issues however require to be clarified.
Absolute numbers of myocarditis and/or pericarditis (M/P) e.g. on figure 2 are a bit misleading. It is necessary to provide also the number of all vaccinated and the percentage of sufferd from M/P. Especially in the context of older vaccines.
Please provide similar adjustments wherever possible, including on figures 4-6.
Please provide, if possible, independent predictors of serious M/P (fatal or life threatening) in patients vaccinated against coronavirus.
Author Response
Absolute numbers of myocarditis and/or pericarditis (M/P) e.g. on figure 2 are a bit misleading. It is necessary to provide also the number of all vaccinated and the percentage of sufferd from M/P. Especially in the context of older vaccines. Please provide similar adjustments wherever possible, including on figures 4-6.
Firstly, we would like to thank the reviewer for careful and thorough reading of this manuscript and for the thoughtful comments and constructive suggestions, which help to improve the quality of this manuscript.
Here are the concerns which can be better addressed.
Answer:
Point 1 We highly appreciated your suggestion to provide among our results also information about all vaccinated and the proportion of subjects which suffered from myocarditis and pericarditis especially in the context of older vaccines. That would be of great interest for our study, unfortunately we could only provide the absolute number of all vaccinated with Covid-19 vaccines from the online website “Our World in Data.org”; indeed, as we pointed out in the conclusion section, we have already used this data source in order to sought the incidence of the study events (myo- and pericarditis).
See Page 10, line 351-354 “indeed, our analysis has shown an incidence of myocarditis following COVID-19 mRNA vaccination equal to 0.38 per 100,000 vaccinated people (95% CI 0.36 – 0.40), in line with a previous literature (0.3 – 0.5 per 100,000 vaccinated people) [27]”.
On the contrary, the absolute number of all vaccinated with older vaccines, such as Smallpox, Anthrax, and Influenza, it is not available, therefore we cannot assess the percentage of the study events related to these vaccines.
Moreover, as we reported in the Methods section, we included in the study all Individual Case Safety Report which described one of the study events (myocarditis, pericarditis, and myopericarditis) regardless of suspected vaccine. Therefore, the Figure n.2 described for each study event the absolute number of myocarditis, pericarditis, and myopericarditis for all the vaccines for which from 1990 to July 2021these events have been reported in the VAERS. In order to better clarify the Figure n.2 we have changed the caption as follows: p.4, lines 135-136 “Distribution of vaccines related to cases of myocarditis (n=1,063), pericarditis (n=741) and myopericarditis (n=37) reported in overall Individual Case Safety Reports (ICSRs).” “Distribution of myocarditis (n=1,063), pericarditis (n=741) and myopericarditis (n=37) by each vaccine for which study events have been reported.”
Please provide, if possible, independent predictors of serious M/P (fatal or life threatening) in patients vaccinated against coronavirus.
Answer:
Point 2 Dear Reviewer, thank you for your comments which give us the opportunity to better explain some key points inherent to the pharmacovigilance studies. Unfortunately, pharmacovigilance databases, such as VAERS, are affected by some limitations, such as the lack of exposure as well as inaccuracy or incompleteness of clinical information (eg. medical conditions). Therefore, we cannot rule out the role of some independent factor especially on fatal or life threatening myo- or pericarditis. Therefore we have added this apect in the discussion section. See p. 8, line 252-255 “However, we cannot rule out the potential role of independent predictors factors like autoimmune disorders as well as systemic inflammatory conditions or beta-blocker therapy in the occurrence of life-threatening and fatal myocarditis and/or pericarditis”.
Reviewer 2 Report
- Authors probe the association of mRNA vaccine and cardiovascular inflammation; pericarditis and myocarditis.
- The subject of the research is interesting and beneficial to researchers in the medical and bioinformatics field.
- The quality of the performed work is distinguished and well-done.
- The article’s flow is harmonious and the narrated sections are well presented.
- The references are relevant and sufficient for the prospective manuscript.
- DECISION: ACCEPTED.

Author Response
Dear Reviewer, thank you for your interest to the topic of our manuscript. We appreciate the time and effort that you have spent to providing positive feedback on our analysis and are grateful for accepting our manuscript.